# International lineages of *Salmonella enterica* serovars isolated from chicken farms, Wakiso District, Uganda

**Takiyah Ball**[1]*, **Daniel Monte**[2], **Awa Aidara-Kane**[3], **Jorge Matheu**[3], **Hongyu Ru**[1], **Siddhartha Thakur**[1], **Francis Ejobi**[4], **Paula Fedorka-Cray**[1]*

**1** Department of Population Health and Pathobiology, North Carolina State University, College of Veterinary Medicine, Raleigh, North Carolina, United States of America, **2** Department of Food and Experimental Nutrition, Faculty of Pharmaceutical Sciences, University of São Paulo, São Paulo, Brazil, **3** Department of Food Safety and Zoonoses, World Health Organization (WHO), Geneva, Switzerland, **4** Department of Biosecurity, Ecosystems & Veterinary Public Health, College of Veterinary Medicine, Animal Resources and Biosecurity, Makerere University, Kampala, Uganda

☯ These authors contributed equally to this work.
* taball@ncsu.edu (TB); pjcray@ncsu.edu (PF-C)

**Data Availability Statement:** The geographical data in this manuscript are available upon request with permission. Due to IRB 17745 restrictions the data was not shown in the manuscript because it shows geographical coordinates of the homes of

## Abstract

The growing occurrence of multidrug-resistant (MDR) *Salmonella enterica* in poultry has been reported with public health concern worldwide. We reported, recently, the occurrence of *Escherichia coli* and *Salmonella enterica* serovars carrying clinically relevant resistance genes in dairy cattle farms in the Wakiso District, Uganda, highlighting an urgent need to monitor food-producing animal environments. Here, we present the prevalence, antimicrobial resistance, and sequence type of 51 *Salmonella* isolates recovered from 379 environmental samples from chicken farms in Uganda. Among the *Salmonella* isolates, 32/51 (62.7%) were resistant to at least one antimicrobial, and 10/51 (19.6%) displayed multiple drug resistance. Through PCR, five replicon plasmids were identified among chicken *Salmonella* isolates including *Inc*FIIS 17/51 (33.3%), *Inc*I1α 12/51 (23.5%), *Inc*P 8/51 (15.7%), *Inc*X1 8/51 (15.7%), and *Inc*X2 1/51 (2.0%). In addition, we identified two additional replicons through WGS (Whole Genome Sequencing; ColpVC and *Inc*FIB). A significant seasonal difference between chicken sampling periods was observed (p = 0.0017). We conclude that MDR *Salmonella* highlights the risks posed to animals and humans. Implementing a robust, integrated surveillance system will aid in monitoring MDR zoonotic threats.

## Introduction

Multidrug-resistant (MDR) *Salmonella enterica* remains a major public health concern as reported in food, animals, humans, and environmental settings, particularly in developing countries. The spread of antimicrobial resistance (AMR) is worldwide [1–6], leading to a high impact on public health and has been deemed a global threat (WHO).

the participants from which the data was collected; therefore identifying their location. In the Wakiso District, Kampala, Uganda commercial farming is primarily at the private homes of the participants. We collected coordinates to identify AMR patterns geographically which is not shown in the manuscript. We understand that this is important data and are willing to share upon request if permission is granted from NC State IRB ethical committee. Please contact the Deb Paxton from the North Carolina State University ethics committee at dapaxton@ncsu.edu as the contact reviewer for this IRB to gain permission for geographical data. Please contact Dr. Paula Fedorka-Cray for available data at pjcray@ncsu. edu.

**Funding:** Funding sources of this project include NC State College of Veterinary Medicine, Makerere College of Veterinary Medicine (FE-Project number SPHQ14-APW-3945) who partially funded this project for collection and culture of the samples and the the World Health Organization as the funder of the grant.

**Competing interests:** The authors have declared that no competing interests exist.

In Uganda, antibiotics, such as tetracyclines and sulfonamides, are increasingly being used and not monitored or regulated in food-producing animals [7]. This practice is well established to select antibiotic-resistant strains that can spread to humans through the food chain. To address this concern and to consider the lack of information regarding AMR in developing countries, Uganda has plans for an integrated national surveillance system for foodborne pathogens using a One Health approach, which is included in their National Action Plan (NAP) for AMR [7].

In Uganda, the poultry production system is divided into two systems, indigenous and exotic flocks. Indigenous chicken, or local birds, make up 88% of the flocks in Uganda, whereas the exotic broilers, kuroilers, and layers make up the rest. There is no current data regarding the total population of poultry in Uganda; however, according to the United Bureau of Statistics (UBOS) in 2008, there were an estimated 52.27 million birds in the country [8]. Hatcheries, which are located in Uganda, are the main source of day-old birds as very few are imported [9]. There is also a lack of information regarding the import and export of live chicken and feed within Uganda as the last census update was conducted in 2005. Feed is supplied to farmers by local feed manufacturers, while a small amount of pre-mixed feed is imported [9]. Commercial poultry in Uganda is primarily kept indoors with screening for ventilation; a small number of chickens are raised at home in the out-of-doors and managed by women and children. Village and backyard production is mainly comprised of free-range poultry [8]. For this study, chickens consisted of broilers, layers, kuroilers, and local (crossbreed) which were housed indoors.

Therefore, we present a cross-sectional study developed in chicken farms in Uganda to investigate the prevalence, AMR, and molecular characterization of *Salmonella enterica* serovars.

## Methods

### Ethical statement

This research was field research on private farms in the Wakiso district of Uganda. There were no field permits required for the sample collection.

We did have an exemption waiver for an IRB for geographical locations used to analyze data of AMR from a geographical standpoint. This data was not used in this manuscript due to IRB ethical concerns. The North Carolina State IRB approval number is 17745.

### Farm description and bacterial isolates

In our previous study, we reported on the phenotypic characterization of *Salmonella* isolates from cattle farms. *Salmonella* isolates were collected from chicken farms in parallel with the collection from cattle farms [5] as part of a cross-sectional study spanning one year. Sampling occurred over two seasons, the rainy season (March to May and September to November) and the dry season (December to February and June to August) [10]. Enrollment in the study occurred through individual contact with producers throughout the Wakiso district. Commercial farms were used in this study located on the west side of Kampala City, Uganda, consisting of rural and small-town farms. Types of chickens on-farm included broilers, layers, kuroilers, and local crossbred chickens, where most farms had two or more types of chickens in production. Most farms had other animals present, either domestic and/or wild, including cattle, horses, pigs, sheep, goats, egrets, turkey, ducks, cats, and dogs.

A total of 20 farms agreed to participate in the study. The first collection was conducted in June (dry seasons), while the second collection was conducted in September (rainy season). A total of 38 farm collections were completed as two farms dropped out of the study in the rainy

season. Ten samples per farm were collected at each visit totaling 379 samples (one farm had nine samples).

Drag swabs (3" x 3" sterile gauze pads) in sterile skim milk was the preferred collection tool (Hardy Diagnostics, Inc., Santa Maria, CA) for farm sampling. The sampling was carried out to ensure maximum sampling of the house floor environment and included inside diagonals, feed and water containers, coops, and outer edge wall-to-wall samples. Swabs were individually placed in a sterile whirl-pak bag; the bag was kept on ice in a cooler prior to transport to the laboratory. Isolation of *Salmonella* was conducted as previously described by Fedorka-Cray et al. [11]. Presumptive-positive *Salmonella* was confirmed using slide agglutination and anti-sera for serogroup determination followed by identification of the *invA* gene (present in all *Salmonella spp.*) by polymerase chain reaction (PCR). All confirmed isolates were frozen in LB broth with 30% glycerol (Thermo Fisher Scientific Inc, Waltham, MA) and stored at -80˚C.

### Antimicrobial resistance and molecular characterization

A total of 51 *Salmonella* were isolated from chicken farms. For analyses, the isolates were retrieved from the -80 frozen stocks, plated on to Tryptic Soy Agar (TSA) with 5% sheep blood (BAP) (Thermo Fisher Scientific Inc, Waltham, MA) and incubated overnight at 37˚C. Antimicrobial resistance testing was done using the National Antimicrobial Resistance Monitoring System (NARMS) Gram-negative panels (Thermo Fisher Scientific Inc, Waltham, MA) as described by Ball et al. [5]. Lysates were prepared by suspending a loopful of well-isolated colonies into 200 µl of molecular grade water and vortexed at maximum speed for several seconds. The suspension was boiled at 100˚C for 10 minutes, centrifuged at 13 X 1000 rpm for 60 seconds, and the supernatant was collected for use as the DNA template. Plasmid detection using PCR was carried out as previously described in Ball et al. [5].

### Whole-genome sequencing

Using the QiAMP commercial kit, DNA extraction was performed (QiAmp tissue, Qiagen, Germany) according to manufacturer's guidelines. Genomic DNA (*n* = 51) were sequenced at a 300-bp paired-end-read using the Nextera XT library preparation kit at the MiSeq platform (Illumina, San Diego, CA). *De novo* assembly was achieved using CLC Genomics Workbench 10.1.1 (Qiagen). Resistome, plasmidome, and multilocus sequence types were identified using multiple public databases such as ResFinder 3.1, PlasmidFinder 2.0, and MLST 2.0, respectively, available from the Center for Genomic Epidemiology (http://genomicepidemiology.org/). Sequence data were deposited in the GenomeTrakr Project.

### Statistical analysis

The prevalence of *Salmonella* was analyzed using WHONET and Microsoft Excel. A logistic regression model was used in SAS® software (SAS® Cary, NC), where season (rainy and dry) served as the factor. Farm was included as a random effect.

## Results

From the 20 farms sampled once during each season, rainy and dry, 379 samples were collected, resulting in 51 positive *Salmonella* isolates. Eight of the 20 farms did not result in a positive sample for *Salmonella* during the study. None of the farms sampled in this study had free-range chickens; all chickens were housed indoors with screening as a source of ventilation. Table 1 displays the results by serotype, AMR phenotype, AMR genotype, and plasmid identification. The 51 *Salmonella* isolates (51/379; 13.5%) belonging to eight different serotypes:

**Table 1. Antimicrobial resistance phenotype and genotype comparison of *Salmonella* from chickens in the Wakiso district of Uganda (n = 51).**

| Farm | Sample ID Biosample # | Season | Serovar | ST | Resistance profile (MIC) | Resistance genes | *gyrA* | *parC* | Plasmids |
|---|---|---|---|---|---|---|---|---|---|
| 1 | SALM-01 SAMN06240035 | Dry | Enteritidis | 11 | Pansusceptible | *strA, strB, aadA1, blaTEM-1B, sul2, sul3, tetA* | none | none | IncFII(S), IncFIB (S), ColpVC |
| 1 | SALM-02 SAMN06240034 | Dry | Enteritidis | 11 | Pansusceptible | Pansusceptible | none | none | IncFII(S), IncFIB (S), ColpVC |
| 1 | SALM-03 SAMN06240033 | Dry | Enteritidis | 11 | Pansusceptible | *sul2* | none | none | IncFII(S), IncFIB (S), ColpVC |
| 1 | SALM-04 SAMN06240032 | Rainy | Typhimurium | 19 | AMP, SOX | *aadA1, blaTEM-1B, qacL, sul3* | none | none | Incl1, IncFII(S), IncFIB (S), ColpVC |
| 1 | SALM-05 SAMN06240031 | Rainy | Enteritidis | 11 | Pansusceptible | Pansusceptible | none | none | IncFII(S), IncFIB (S), ColpVC |
| 1 | SALM-06 SAMN06240030 | Rainy | Typhimurium | 19 | AMP, SOX | *aadA1, blaTEM-1B, qacL, sul3* | none | none | Incl1, IncFII(S), IncFIB (S), ColpVC |
| 1 | SALM-07 SAMN06240029 | Rainy | Enteritidis | 11 | Pansusceptible | Pansusceptible | none | none | IncFII(S), IncFIB (S), ColpVC |
| 3 | SALM-08 SAMN06240028 | Dry | Kentucky | 198 | AMP, CIP, NAL, STR, SOX, TCY, SXT | *aadA1, aph(6)-Id, strA, strB, blaTEM-1B, dfrA14, qacL, sul2, sul3, tet(A)* | S83F/ D87N | S80I | ColpVC, Incl1 |
| 3 | SALM-09 SAMN06240027 | Dry | Kentucky | 198 | CIP, NAL | Pansusceptible | S83F/ D87N | S80I | ColpVC |
| 3 | SALM-10 SAMN06240026 | Dry | Kentucky | 198 | CIP, NAL, STR, SOX, TCY | *aph(3″)-Ib, aph(6)-Id, strA, strB, sul2, tet (A)* | S83F/ D87N | S80I | ColpVC |
| 3 | SALM-11 SAMN06240025 | Dry | Kentucky | 198 | CIP, NAL, STR, SOX, TCY | *aph(3″)-Ib, aph(6)-Id, strA, strB, sul2, tet (A)* | S83F/ D87N | S80I | ColpVC |
| 3 | SALM-12 SAMN06240024 | Rainy | Kentucky | 198 | CIP, NAL, STR, SOX, TCY | *aph(3″)-Ib, aph(6)-Id, strA, strB, sul2, tet (A)* | S83F/ D87N | S80I | ColpVC |
| 3 | SALM-13 SAMN06240023 | Rainy | Kentucky | 198 | CIP, NAL, STR, SOX, TCY | *aph(3″)-Ib, aph(6)-Id, strA, strB, sul2, tet (A)* | S83F/ D87N | S80I | ColpVC |
| 3 | SALM-14 SAMN06240022 | Rainy | Kentucky | 198 | CIP, NAL, STR, SOX, TCY | *aph(3″)-Ib, aph(6)-Id, strA, strB, sul2, tet (A)* | S83F/ D87N | S80I | ColpVC |
| 3 | SALM-15 SAMN06240021 | Rainy | Kentucky | 198 | CIP, NAL, STR, SOX, TCY | *aph(3″)-Ib, aph(6)-Id, strA, strB, sul2, tet (A)* | S83F/ D87N | S80I | ColpVC |
| 3 | SALM-16 SAMN06240020 | Rainy | Kentucky | 198 | CHL, CIP, NAL, STR, SOX, TCY, SXT | *qnrS1, aadA1, aadA2, aph(6)-Id, strA, strB, cmlA1, dfrA14, sul2, sul3, tet(A)* | S83F/ D87N | S80I | ColpVC, Incl1 |
| 3 | SALM-17 SAMN06240019 | Rainy | Kentucky | 198 | CIP, NAL, STR, SOX, TCY | *strA, strB, sul2, tetA* | none | none | ColpVC |
| 3 | SALM-18 SAMN06240018 | Rainy | Kentucky | 198 | AMP, CIP, NAL, SOX | *aadA1, blaTEM-1B, sul3* | S83F/ D87N | S80I | ColpVC, Incl1 |
| 4 | SALM-19 SAMN06240017 | Dry | Zanzibar | 466 | TCY | *tetA* | none | none | Incl1 |
| 4 | SALM-20 SAMN06240016 | Dry | Zanzibar | 466 | TCY | *tetA* | none | none | Incl1 |
| 4 | SALM-21 SAMN06240015 | Dry | Zanzibar | 466 | TCY | *tetA* | none | none | ColpVC, Incl1 |
| 4 | SALM-22 SAMN06240014 | Dry | Zanzibar | 466 | TCY | *tetA* | none | none | Incl1 |
| 4 | SALM-23 SAMN06240013 | Dry | Zanzibar | 466 | TCY | *tetA* | none | none | ColpVC, Incl1 |
| 4 | SALM-24 SAMN06240012 | Dry | Zanzibar | 466 | Pansusceptible | Pansusceptible | none | none | ColpVC |
| 4 | SALM-25 SAMN06240092 | Dry | Zanzibar | 466 | TCY | *tetA* | none | none | Incl1 |

(*Continued*)

**Table 1.** (*Continued*)

| Farm | Sample ID Biosample # | Season | Serovar | ST | Resistance profile (MIC) | Resistance genes | *gyrA* | *parC* | Plasmids |
|------|----------------------|--------|---------|----|--------------------------|------------------|--------|--------|----------|
| 4 | SALM-26 SAMN06240091 | Rainy | Enteritidis | 11 | Pansusceptible | Pansusceptible | none | none | IncFII(S), IncFIB (S) |
| 4 | SALM-27 SAMN06240090 | Rainy | Zanzibar | 466 | TCY | *tetA* | none | none | IncI1 |
| 4 | SALM-28 SAMN06240089 | Rainy | Enteritidis | 11 | Pansusceptible | Pansusceptible | none | none | IncFII(S), IncFIB (S) |
| 4 | SALM-29 SAMN06240088 | Rainy | Enteritidis | 11 | Pansusceptible | Pansusceptible | none | none | IncFII(S), IncFIB (S) |
| 4 | SALM-30 SAMN06240087 | Rainy | Enteritidis | 11 | Pansusceptible | Pansusceptible | none | none | IncFII(S), IncFIB (S) |
| 4 | SALM-31 SAMN06240086 | Rainy | Enteritidis | 11 | Pansusceptible | Pansusceptible | none | none | IncFII(S), IncFIB (S) |
| 4 | SALM-32 SAMN06240085 | Rainy | Enteritidis | 11 | Pansusceptible | Pansusceptible | none | none | IncFII(S), IncFIB (S) |
| 5 | SALM-33 SAMN06240084 | Rainy | Enteritidis | 11 | Pansusceptible | Pansusceptible | none | none | IncFII(S), IncFIB (S) |
| 8 | SALM-34 SAMN06240083 | Rainy | Virchow | 16 | NAL, TCY | *tetA* | S83Y | none | none |
| 8 | SALM-35 SAMN06240082 | Rainy | Virchow | 16 | NAL, TCY | *tetA* | S83Y | none | none |
| 8 | SALM-36 SAMN06240081 | Rainy | Virchow | 16 | NAL, TCY | *tetA* | S83Y | none | none |
| 8 | SALM-37 SAMN06238262 | Rainy | Virchow | 16 | NAL, TCY | *tetA* | S83Y | none | none |
| 8 | SALM-38 SAMN06238261 | Rainy | Virchow | 16 | NAL, TCY | *tetA* | S83Y | none | none |
| 8 | SALM-39 SAMN06238260 | Rainy | Virchow | 16 | NAL, TCY | *tetA* | S83Y | none | none |
| 8 | SALM-40 SAMN06238259 | Rainy | Virchow | 16 | NAL, TCY | *tetA* | S83Y | none | none |
| 9 | SALM-41 SAMN06238258 | Rainy | Enteritidis | 11 | Pansusceptible | Pansusceptible | none | none | IncFII(S), IncFIB (S), Col440I |
| 9 | SALM-42 SAMN06238257 | Rainy | Enteritidis | 11 | Pansusceptible | Pansusceptible | none | none | none |
| 9 | SALM-43 SAMN06238276 | Rainy | Enteritidis | 11 | Pansusceptible | Pansusceptible | none | none | IncFII(S), IncFIB (S), Col440I |
| 10 | SALM-44 SAMN06238275 | Dry | 42:r:- | 1208 | STR | Pansusceptible | none | none | Col440I |
| 14 | SALM-45 SAMN06238274 | Rainy | Newport | 166 | NAL, TCY | *qnrS1*, *tetA* | none | none | IncX2 |
| 14 | SALM-46 SAMN06238273 | Rainy | 42:r:- | 1208 | STR | *Pansusceptible* | none | none | none |
| 15 | SALM-47 SAMN06238272 | Dry | Barranquilla | 3807 | Pansusceptible | Pansusceptible | none | none | none |
| 17 | SALM-48 SAMN06238271 | Rainy | Virchow | 16 | NAL, TCY | *tetA* | S83Y | none | none |
| 17 | SALM-49 SAMN06238270 | Rainy | Enteritidis | 11 | Pansusceptible | Pansusceptible | none | none | IncFII(S), IncFIB (S) |
| 18 | SALM-50 SAMN06238269 | Dry | Newport | 46 | Pansusceptible | Pansusceptible | none | none | none |

(*Continued*)

**Table 1.** (Continued)

| Farm | Sample ID Biosample # | Season | Serovar | ST | Resistance profile (MIC) | Resistance genes | *gyrA* | *parC* | Plasmids |
|------|------------------------|--------|---------|-----|--------------------------|------------------|--------|--------|----------|
| 20 | SALM-51 SAMN06238268 | Rainy | 42:r:- | 1208 | STR | Pansusceptible | none | none | none |

Antibiotics: AMC = Amoxicillin-Clavulanic Acid, AMP = Ampicillin, AZM = Azithromycin, FOX = Cefoxitin, TIO = Ceftiofur, CRO = Ceftriaxone, CHL = Chloramphenicol, CIP = Ciprofloxacin, GEN = Gentamicin, NAL = Nalidixic Acid, STR = Streptomycin, SOX = Sulfisoxazole, TCY = Tetracycline, SXT = Trimethoprim-Sulfamethoxazole

Farms that are not displayed were negative for *Salmonella* (2,6,7,11,12,13,16,19). All farms that show a negative sign in the table were negative for *Salmonella* for that particular season. Farm 11 was negative for *Salmonella* for the dry season and did not participate for the rainy season; therefore, both are not shown in the table. Farm 15 only participated in the dry season, as shown in the table.

*Salmonella* serovar Enteritidis (31.3%); *S.* Kentucky (21.6%); *S.* Zanzibar and *S.* Virchow (15.7%); *S.* Newport and *S.* serovar 42:r:- (5.88%), *S.* Typhimurium (4%) and *S.* Barranquilla at (2.0%). The overall prevalence of *Salmonella* was higher in the rainy season (p = 0.0017). No interaction between serotype and season was observed.

The isolates displayed resistance to eight antimicrobials including tetracycline (51%), nalidixic acid (37.3%), sulfisoxazole (23.5%), ciprofloxacin (21.6%), streptomycin (13.7%), ampicillin (7.8%), sulfamethoxazole (3.9%), and chloramphenicol (2%). Phenotypically, all *Salmonella* Enteritidis were pan-susceptible, and all except one *Salmonella* Kentucky were MDR isolates. No interaction was observed between serotype and season (Table 2).

Table 3 and Table 4 display the AMR phenotypes by class of antibiotics as well as the frequency (%) of resistance patterns. Ten isolates (all of which are *Salmonella* Kentucky) displayed MDR (resistant to three or more classes), as seen in Table 3. Resistance to both nalidixic acid and tetracycline only occurred within *Salmonella* serovars Virchow and Newport. Other patterns observed included TCY (*S.* Zanzibar), STR (*S.* 42:r-), and AMP-SOX (*S.* Typhimurium) resistance. All other patterns were observed among *S.* Kentucky isolates.

**Table 2. Serotype distribution on farms by season.**

| Farm ID | Season | |
|---------|--------|--------|
| | **Dry** | **Rainy** |
| 1 | *S.* Enteritidis (n = 2), *S.* Typhimurium (n = 2) | *S.* Enteritidis (n = 3) |
| 3 | *S.* Kentucky (n = 7) | *S.* Kentucky (n = 4) |
| 4 | *S.* Zanzibar (n = 1), *S.* Enteritidis (n = 6) | *S.* Zanzibar (n = 7) |
| 5 | *S.* Enteritidis (n = 1) | - |
| 8 | - | *S.* Virchow (n = 7) |
| 9 | - | *S.* Enteritidis (n = 3) |
| 10 | *S.* 42:r- (n = 1) | - |
| 14 | - | *S.* 42:r- (n = 1), *S.* Newport (n = 1) |
| 15 | *S.* Barranquilla (n = 1) | ND |
| 17 | - | *S.* Virchow (n = 1), *S.* Enteritidis (n = 1) |
| 18 | *S.* Newport (n = 1) | - |
| 20 | - | *S.* 42:r- (n = 1) |

Farms that are not displayed were negative for *Salmonella* (2,6,7,11,12,13,16,19). All farms that show a negative sign in the table were negative for *Salmonella* for that particular season. Farm 11 was negative for *Salmonella* for the dry season and did not participate for the rainy season; therefore, both are not shown in the table. Farm 15 only participated in the dry season, as shown in the table.

**Table 3. MDR resistance of *Salmonella* from chicken (n = 51).**

| Resistance Pattern | N (%) |
|---|---|
| No Resistance Detected | 19 (37.3) |
| Resistance = 1 CLSI Class[1] | 11 (21.6) |
| Resistance = 2 CLSI Classes[1] | 11 (21.6) |
| Resistance = 3 CLSI Classes[1] | 1 (2.0) |
| Resistance = 4 CLSI Classes[1] | 7 (13.7) |
| Resistance = 5 CLSI Classes[1] | 2 (3.9) |

Clinical and Laboratory Standards Institute Class[1]: Antibiotic class including penicillin

Whole-genome sequencing analysis revealed the presence of resistance genes to tetracycline [*tetA*; 53%], sulfonamides [*sul2* (21.5%); *sul3* (11.7%)], streptomycin [*strA* (19.6%); *strB* (19.6%)], aminoglycosides [*aph(6)-Id* (15.6%); *aph(3″)-Ib* (11.7%); *aadA1* (11.7%); *aadA2* (2%)], β-lactams [*bla*$_{TEM-1B}$; 9.8%], quaternary ammonium [*qacL*; 5.8%], quinolones [*qnrS1*; 5.8%] and trimethoprim [*dfrA14*; 4%]. Genes were noted as quinolone resistance determining regions (QRDR) with point mutations in *gyrA* and *parC* (Table 1). Ten isolates (19.6%) showed a double amino acid mutation in GyrA (GyrA-S83F-D87N), whereas eight isolates (15.6%) showed a single amino acid substitution of serine to tyrosine at codon 83. For QRDR in *parC* (n = 10; 19.6%), only one substitution in serine to isoleucine at codon 80 was observed. Sequencing identified six plasmids. *Inc*FII(S)-*Inc*FIB (S)-ColpVC were most common in *S.* Enteritidis; *Inc*l1-ColpVC in *S.* Kentucky and *S.* Zanzibar; *Inc*X2 in *S.* Newport; *Inc*l1-*Inc*FII (S)-*Inc*FIB (S)-ColpVC in *S.* Typhimurium and Col440I in *S.* serovar 42:r:-. Nine sequence types (ST), namely ST11, ST198, ST466, ST16, ST166, ST46, ST19, ST1208, and ST3807 were associated with *S.* Enteritidis, *S.* Kentucky, *S.* Zanzibar, *S.* Virchow, *S.* Newport, *S.* Newport, *S.* Typhimurium, *S.* serovar 42:r:- and *S.* Barranquilla, respectively. Five of the 28 plasmids that were screened through PCR were observed in multiple isolates: *Inc*FIIS (17/51; 33.3%), *Inc*I1α (12/51; 23.5%), *Inc*P (8/51; 15.7%), 193 *Inc*X1 (8/51; 15.7%), and *Inc*X2 (1/51; 2.0%). After analyzing the WGS sequences for plasmids, 12 isolates were found to harbor *Inc*I1α, with seven of the 12 having an additional plasmid (ColpVC) that was not detected by PCR (ColpVC was not included in the PCR kit) and two with *Inc*FIIS plasmid. Seventeen isolates carried the *Inc*FIIS

**Table 4. Top resistance patterns for *Salmonella* from chicken (n = 51).**

| Resistance pattern | N (%) |
|---|---|
| NAL TCY | 9 (17.6) |
| TCY | 7 (13.7) |
| CIP NAL STR SOX TCY | 7 (13.7) |
| STR | 3 (5.9) |
| AMP SOX | 2 (3.9) |
| CIP NAL | 1 (2.0) |
| AMP CIP NAL SOX | 1 (2.0) |
| CHL CIP NAL STR SOX TCY SXT | 1 (2.0) |
| AMP CIP NAL STR SOX TCY SXT | 1 (2.0) |

**Antibiotics:** AMC = Amoxicillin-Clavulanic Acid, AMP = Ampicillin, AZM = Azithromycin, FOX = Cefoxitin, TIO = Ceftiofur, CRO = Ceftriaxone, CHL = Chloramphenicol, CIP = Ciprofloxacin, GEN = Gentamicin, NAL = Nalidixic Acid, STR = Streptomycin, SOX = Sulfisoxazole, TCY = Tetracycline, SXT = Trimethoprim-Sulfamethoxazole

plasmid. These same 17 isolates also presented *Inc*FIB (S) plasmids, and ColpVC and Col4401 were identified in seven and two isolates, respectively. *Inc*X2 and *Inc*P were not identified in the WGS analysis and by PCR. PCR did not detect plasmids in ten isolates, but WGS detected ColpVC in nine isolates and Col4401 in one isolate. *Inc*FIIS was the most common plasmid identified at 33.3% (17/51). Overall, it was seen how the use of WGS presented a more robust and accurate data analysis for resistance genes present in the isolates. Phenotypic data will not always allow for a good representation of what genes are present as genotypic data.

Based on the output provided for this study, there was a significance (p = 0.0017) seen during the rainy seasons as compared to the dry with a higher presence of positive *Salmonella*.

## Discussion

The percent prevalence of *Salmonella* (13.5%) in this study highlights the potential risk to humans in Ugandan households, particularly those engaging in poultry production. There is a lack of reports on the prevalence of *Salmonella* on farm; the percentage reported in this study is slightly higher than the 11% reported by Afema et al. [12] and comparable to the farms in Nigeria at 2–26%[13]]. As the majority of chickens from the farms in this study end up for sale at the live market, the prevalence is likely in concordance with what is seen on farm. This heightens the concern that food-animals are a possible source of *Salmonella* for Ugandan consumers, regardless of AMR status, further highlighting the need for control of zoonotic pathogens, including *Salmonella*.

We also learned that there was a seasonal effect associated with the recovery of *Salmonella*. Uganda typically has a rainy season that occurs between March to May and September to November [10]. Recovery of *Salmonella* was higher during the rainy season, and the use of screening does not allow for temperature control. Therefore, it is likely that the higher humidity and moisture allowed for better dispersal or survival of *Salmonella* as observed for several bacterial species in poultry[14]. Further, grass is not commonly seen around production buildings and during a rain event, as the environment is mostly mud. It is also possible that human traffic during daily chores resulted in higher traffic of *Salmonella* into the facility. Additional environmental studies are warranted.

Comparable to the United States (US) [15], *Salmonella* serovars Enteritidis and Kentucky were most often recovered from chicken samples. Serovar Kentucky has previously been reported in Uganda in humans, poultry, and the environment [12] However, there were no similarities between *Salmonella* serovars reported in humans compared with the serovars observed in our study. Afema et al. [12], reported that *Salmonella* Haifa was most commonly seen in samples collected from wastewater treatment plants in Kampala city, along with *S.* Stanleyville, *S.* Kentucky, *S.* Heidelberg, and *S.* 42r:- rounding out the top five; however, *Salmonella* Enteritidis was not detected in human samples from this study [12]. While there are similarities between serovars from the wastewater treatment plants, the source of the isolates is unknown. It should be noted that most of the housing outside of Kampala proper does not include indoor plumbing and outhouses are prevalent. Further, at the live market, particularly in small villages, flush gutters are used and animals are dressed on-site with waste commonly ending up in the gutter. The gutters are also used for dumping wash water, garbage, and other waste as well for the passage of human waste. Environmental studies would be quite complex, and multiple factors would need to be controlled for. This highlights the complexity and crucial component of the environment in determining the source of pathogens in Uganda.

Approximately 38% of the isolates were resistant to two or more classes of antibiotics, including two isolates that were resistant to seven antimicrobials. Interestingly, there was no resistance to third-generation cephalosporins. This was also noted from cattle samples, as

described in our previous report [5]. As third-generation cephalosporins are the treatment of choice when indicated for salmonellosis, surveillance for emerging resistance is warranted and may aid in identifying the source of infection.

The *Salmonella* serovar Kentucky isolates were resistant to over five (ciprofloxacin, nalidixic acid, streptomycin, sulfisoxazole, and tetracycline) or seven (chloramphenicol, ampicillin, ciprofloxacin, nalidixic acid, streptomycin, sulfisoxazole, tetracycline, and trimethoprim-sulfamethoxazole) antibiotics. All *S*. Kentucky isolates were resistant to ciprofloxacin, and all originated from one farm; however, the only antibiotic used on that farm was oxytetracycline with water as the route of administration. The source of the ciprofloxacin resistance is unknown as it is not used in poultry production; the only other animal on this farm were dogs. Since the early 2000s, ciprofloxacin resistance in *Salmonella* serovar Kentucky has been on the rise, especially from travelers to northern and eastern Africa [16]. Rickert-Hartman et al. [16] found that 9% of the *Salmonella* serovar Kentucky isolates from travelers were ciprofloxacin-resistant. Poultry was thought to be a reservoir for these resistant strains [16, 17]. Ciprofloxacin-resistant *S*. Kentucky was attributed to illness in seven people and one death in the US after traveling from India [16]. In this regard, the emergence of *S*. Kentucky ST198, which is resistant to a number of critically important antibiotics, poses a major threat to public health worldwide since it is highly drug-resistant [18] and has been reported from different sources including retail chicken carcasses [19]. The presence of the mutation can be useful for tracking the pandemic ciprofloxacin-resistant *S*. Kentucky strain ST198 from geographically distinct regions [18].

Other serotypes exhibiting MDR includes *Salmonella* Newport, which has recently been reported by the Centers for Disease Control and Prevention as having ciprofloxacin and azithromycin resistance in the US; the origin was soft cheese and beef from the US and Mexico, respectively [20]. Globally, MDR has also been reported for DT104 *S*. Typhimurium [21, 22].

The AMR field is moving to utilize WGS for detecting resistant genes worldwide. We sequenced all isolates to identify resistance genes and compare them to the observed AMR phenotype. With WGS, the β-lactamase gene $bla_{TEM-1B}$ was identified in five isolates that were not identified by PCR. In previous studies [23], discrepancies were also seen between phenotypic resistance patterns and genotypic analysis using WGS. It was reported that a MIC might not reach the breakpoint even though resistance genes were present [23]. In some cases, in this study, the gene was not present but was expressed phenotypically, which is not typically seen. Little research is done as to why this happens and will need further investigation.

WGS was also used to detect plasmids and compare the results with PCR. All results were in concordance with PCR and WGS, except for *S*. Virchow isolates. As stated above, AMR genes were not present by WGS but were observed phenotypically for *S*. Virchow isolates contained *Inc*P and *Inc*X1 plasmids. As with the AMR genes, false positives may explain this phenomenon, but further testing needs to elucidate the differences between the PCR and WGS results.

In this study, *Inc*FIIS was the most common plasmid identified (33.3% (17/51)). Studies have shown that bacterial isolates containing $bla_{CTX-M-1}$, harbor the *Inc*FIIS plasmid along with other incompatibility plasmids [24]. *Inc*1 plasmids are known to be distributed throughout many serotypes of *Salmonella* and predominate in both *E.coli* and *Salmonella* [25–27]. *Inc*1α was observed among *Salmonella* serovars Zanzibar, Kentucky, and Typhimurium. *Inc*P and *Inc*X1 were the next most common plasmids detected by PCR. Both were present in the *Salmonella* serovar Virchow isolates. It has been reported that *Inc*P plasmids can spread via conjugative transfer and that they code for a broad range of antimicrobial resistance. *Inc*P is highly likely to be found in manure, wastewater, and soil [28]. *Inc*X1 is commonly found as a narrow host-range plasmid in Enterobacteriaceae, also spreading to other bacteria via conjugative transfer [29].

Although traditional tools have been considered the gold standard to study *Salmonella*, WGS has been applied as an alternative in providing more detailed and accurate data. In this regard, WGS identifies antimicrobial resistance profile, MLST, and evolutionary groupings that could precisely determine the differences between *Salmonella* strains. We observed that the main drivers for characterization analysis were serotype, sequence types, and resistance profile. These isolates were clustered together by these characteristics and not by a period of isolation, source, or geographic location. To endorse these results, we have done pulsed-field gel electrophoreses (results not included), which are in agreement with the WGS results. Our study shows how WGS inspection constitutes a useful means to characterize *Salmonella* isolates.

## Conclusion

In summary, we present in this study eight *Salmonella* enterica serovars displaying resistance to clinically important antibiotics. Of these, the presence of international lineages as ciprofloxacin-resistant *S*. Kentucky sequence type 198 in chicken farms presents public concern given that fluoroquinolones are the first treatment choice. Our findings suggest the occurrence of epidemic dissemination of resistant serovars, adding valuable information and justification for establishing a robust epidemiological One Health integrated surveillance program in Uganda. Therefore, these results may encourage additional genomic surveillance studies in this region to aid the development of mitigation strategies and to limit the global distribution of these multi-drug resistant *Salmonella enterica* isolates.

## Acknowledgments

We want to acknowledge our colleagues at Makerere University Dr. Eddie Wampande, Sarah Tegule, Samuel Maling, David Apollo Munanura, Allan Odeke, Disan Muhangazi, Mark Ogal, Mutumba Paul, and Elizabeth Basemera; Our colleagues at the NCSU, College of Veterinary Medicine Diagnostic Laboratory, Dr. Megan Jacob.

## Author Contributions

**Conceptualization:** Takiyah Ball, Awa Aidara-Kane, Jorge Matheu, Francis Ejobi, Paula Fedorka-Cray.

**Data curation:** Takiyah Ball, Daniel Monte.

**Formal analysis:** Takiyah Ball, Daniel Monte, Hongyu Ru.

**Funding acquisition:** Takiyah Ball, Awa Aidara-Kane, Jorge Matheu, Paula Fedorka-Cray.

**Investigation:** Takiyah Ball, Jorge Matheu, Francis Ejobi.

**Methodology:** Takiyah Ball, Jorge Matheu, Siddhartha Thakur, Paula Fedorka-Cray.

**Project administration:** Takiyah Ball, Awa Aidara-Kane, Francis Ejobi, Paula Fedorka-Cray.

**Resources:** Takiyah Ball, Jorge Matheu, Francis Ejobi, Paula Fedorka-Cray.

**Software:** Takiyah Ball.

**Supervision:** Takiyah Ball, Jorge Matheu, Francis Ejobi, Paula Fedorka-Cray.

**Validation:** Takiyah Ball.

**Visualization:** Takiyah Ball.

**Writing – original draft:** Takiyah Ball, Daniel Monte.

**Writing – review & editing:** Takiyah Ball, Daniel Monte, Jorge Matheu, Francis Ejobi, Paula Fedorka-Cray.

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
