## [Decision Letter · Decision Letter 0]

19 Aug 2019

PONE-D-19-19750

International lineages of Salmonella enterica serovars isolated from chicken farms, Wakiso District, Uganda

PLOS ONE

Dear Dr. Ball,

Thank you for submitting your manuscript to PLOS ONE. After careful consideration, we feel that it has merit but does not fully meet PLOS ONE’s publication criteria as it currently stands. Therefore, we invite you to submit a revised version of the manuscript that addresses the points raised during the review process.

We would appreciate receiving your revised manuscript by Oct 03 2019 11:59PM. To enhance the reproducibility of your results, we recommend that if applicable you deposit your laboratory protocols in protocols.io, where a protocol can be assigned its own identifier (DOI) such that it can be cited independently in the future. For instructions see: http://journals.plos.org/plosone/s/submission-guidelines#loc-laboratory-protocols

We look forward to receiving your revised manuscript.

Kind regards,

Feng Gao

Academic Editor

PLOS ONE

**Journal Requirements:**

2. In your Methods section, please provide additional location information, including geographic coordinates for the data set if available.

3. In your Methods section, please describe whether you received permission from farm owners to carry out the work in this study.

4.To comply with PLOS ONE submissions requirements for field studies, please provide the following information in the Methods section of the manuscript and in the “Ethics Statement” field of the submission form (via “Edit Submission”):

a) Provide the name of the authority who issued the permission for each location (for example, the authority responsible for a national park or other protected area of land or sea, the relevant regulatory body concerned with protection of wildlife, etc.).

If the study was carried out on private land, please confirm that the owner of the land gave permission to conduct the study on this site.

b) For any locations/activities for which specific permission was not required, please

- i. state clearly that no specific permissions were required for these locations/activities, and provide details on why this is the case

- ii. confirm that the field studies did not involve endangered or protected species

c) For vertebrate studies only, please provide the following additional information:

- i. Full details of collection and sampling methods, including method of sacrifice if applicable

- ii. State whether the vertebrate work was approved by an Institutional Animal Care and Use Committee (IACUC) or equivalent animal ethics committee. If no approval was obtained, please explain why it was not required.

- iii. State clearly whether all sampling procedures and/or experimental manipulations were reviewed or specifically approved as part of obtaining the field permit.

For more information about PLOS ONE submissions requirements for field studies, please refer to http://journals.plos.org/plosone/s/submission-guidelines#loc-animal-research.

**Comments to the Author**

1. Is the manuscript technically sound, and do the data support the conclusions?

Reviewer #1: Yes

Reviewer #2: Partly

Reviewer #3: Yes

Reviewer #4: Yes

Reviewer #5: Yes

Reviewer #6: Partly

2. Has the statistical analysis been performed appropriately and rigorously? 

Reviewer #1: Yes

Reviewer #2: N/A

Reviewer #3: Yes

Reviewer #4: N/A

Reviewer #5: I Don't Know

Reviewer #6: N/A

3. Have the authors made all data underlying the findings in their manuscript fully available?

Reviewer #1: Yes

Reviewer #2: Yes

Reviewer #3: Yes

Reviewer #4: Yes

Reviewer #5: No

Reviewer #6: Yes

4. Is the manuscript presented in an intelligible fashion and written in standard English?

Reviewer #1: Yes

Reviewer #2: Yes

Reviewer #3: No

Reviewer #4: Yes

Reviewer #5: No

Reviewer #6: Yes

5. Review Comments to the Author

Reviewer #1: Review of International lineages of Salmonella enterica serovars isolated from chicken farms, Wakiso District, Uganda, by Ball et al.

In this manuscript, 51 isolates of Salmonella from chicken farms were characterized with respect to antimicrobial resistance, PCR testing and whole genome sequencing. The work seems to have been carried out carefully, but some details are missing in the materials and methods. Logistic regression was carried out with season as main factor, but it is not clear if only the total numbers of Salmonella strains were considered as response factor or also different genotypes and antimicrobial resistance factors. The main problem I have with the current manuscript is that differences were observed between seasons for total Salmonella counts, but it is not clear if the genotypic and phenotypic compositions of the Salmonella isolates were different between the rainy and dry seasons.

I made detailed comments in the manuscript itself; a scanned copy will be attached.

Title:

Replace Review of International Lineages by: Internationally recognized lineages.

This is not a review, and the lineages are all from Uganda.

Abstract:

Generally ok, but see comments in the attachment.

Introduction:

OK, but add a paragraph about the methods of chicken raising in Uganda. Are the farms large-scale indoor chicken coops or are the chickens outdoors? This is important to be able to understand the differences between the rainy and dry seasons. Mention also about the movement of Salmonella between habitats (soil, feed, water, chickens, manure, floor, walls etc.) (Semenov et al., 2010).

Semenov, A.M., Kupriyanov, A.A. and van Bruggen, A.H.C. 2010. Transfer of enteric pathogens to successive habitats as part of microbial cycles. Microb. Ecol. 60: 239-249.

Methodology:

The rainy season and dry season overlap (line 64). Was the period of overlap between seasons avoided during sampling? Or were some samples considered in both categories?

The isolation method by Fedorka-Cray et al. was used; this is a classical method with the possibility that some colonies were considered Salmonella, but actually were not. How was the identity of all these isolates checked? By PCR? Which primers were used? That needs to be mentioned. See also Klerks et al., 2006.

Klerks, M.M., van Bruggen, A.H.C., Zijlstra, C., Donnikov M., and de Vos, R. 2006. Comparison of methods of extracting Salmonella enterica serovar Enteritidis DNA from environmental substrates and quantification of organisms by using a general internal procedural control. Appl. Environ. Microbiol. 72: 3879-3886.

Results:

It is important to understand the differences between the rainy and dry seasons. I suggest indicating in Table 1 which samples were collected in each of the seasons. Then you can mention which serotypes occurred in the rainy season and which in the dry season. Now, the numbers may be too low in the dry season, but then it is important to mention that these are preliminary data, and that additional research would be needed to distinguish between seasons.

The outcome of the statistical analysis has not been presented in detail. I expect a multivariate graph, including all observations. Hopefully those will group out into two ‘clouds’, one for the rainy season and one for the dry season.

Discussion:

To better understand the results, in particular the differences between dry and rainy seasons, you need to refer back to the way chickens are raised in Uganda (that’s why you need to add a paragraph about this in the introduction, so that you can refer to it in the discussion).

Which serotypes were endemic and which were introduced? In the conclusion you mention endemic dissemination. In the rest of the manuscript, nothing was mentioned about dissemination. Are chickens imported or exported? Is the feed imported or exported? It is not really clear where all these serotypes and multi-drug-resistances came from. Or did the drug resistance develop in Uganda? It is important to give more general information about chicken production and the possible routes of dissemination of Salmonella.

Reviewer #2: The article by Ball et al. tackles the vital subject of antimicrobial resistance food-producing animals in a region from where very little is known regarding this subject. It is a welcome addition to the body of knowledge, although the article is missing information and data analysis.

1- The farms a poorly described both, in a geographical sense and relation to each other. Furthermore, the type of production, size, and information on different types of animals on the farm are not mentioned but are necessary.

2- Spatial and temporal information on the tables is missing, and it is barely described in the results, and not discussed.

3- The presence of resistance genes without phenotypic testing is an indication of resistance, but the isolates cannot be described as resistant as it is the case in the discussion.

5- Information on the quality of the sequencing obtained for each isolate should be documented. If they met Genometrackr standards, please estate this fact, and their current minimum.

4-Exact information on the bioinformatics pipeline used to analyze the samples Is crucial for each isolate. Where different databases used every time? Where there any differences between the databases?

Reviewer #3: This paper describes a cross sectional study of Salmonella in poultry farms in Uganda conducted over the dry and rainy seasons. The authors undertook antibiogram phenotyping of the isolate collection and used whole genome sequence analysis to determine the serotype, AMR genotype, plasmid replicon type and in some cases sequence type of the isolates. A high proportion of isolates were found to be MDR and to contain plasmids of significant impact to human health and alarmingly, fluoroquinolone resistance was very high (21.6%). Whilst the study design is excellent, the techniques used appropriate and the results are significant, the paper is let down by being poorly written in parts. Also, it is not stated whether the study population are broilers or layers (or mixed?). I would also like to see more discussion about the very high rate of fluoroquinolone resistance and its relationship to antimicrobial use on farm. The absence of resistance to third generation cephalosporins is also an important finding.

Specific comments below:

Line 45-47: This is a poor sentence that needs rewriting

Methods: Were these broiler farms or egg-producing farms or a combination of the two? This is a very important distinction due to the restrictions on which antimicrobials can be used in each system.

Line 113: Tetracycline spelt incorrectly. It is not the AMR phenotype displaying resistance to eight antimicrobials but the isolate collection.

Line 120: Other than acquiring resistance genes does not make sense, This should be “Point mutations identified in the QRDR of fluoroquinolone target genes included……”

The section from Line 120 to Line 127 needs a careful rewrite as the sentence construction is poor.

Line 128 to 130 is poorly written

Reviewer #4: Multiple drug resistance (MDR) foodborne pathogens have been reported from multiple sources, especially from poultry worldwide. In this article, Ball et al. characterized 51 Salmonella enterica isolates, isolated from 400 environmental samples collected form poultry farms in Uganda by using whole genome sequencing (WGS). In addition, they also identified replicon plasmid through PCR. Among 51 isolates, 32 isolates were resistant to at least one antimicrobial and 10 isolates were assigned as MDR. Ball et al. also found a significant correlation between prevalence of Salmonella and seasonal difference. This article is underlining the importance of the surveillance systems, using One Health approach, to control MDR foodborne pathogens world wide. However, there are some minor issues that should be addressed and clarified before prior to publication.

Minor issues:

- Footnote, describing abbreviations of antimicrobials, should be added to Table 1. In addition, to describe temporal and spatial relatedness of isolates, new columns, representing where and when isolates were collected, might help.

- For plasmid description, there are some inconsistent results, obtained from WGS and PCR. These results should be discussed in details.

- For antimicrobial resistance profiling, there are also some inconsistent results. For examples, some isolates (i.e., SALM-9, SALM-1) represented resistance by using MIC, but no gene, related to this phenotype, was found from WGS or vice versa. These results should be discussed in details.

- In general, more discussion should be added.

Reviewer #5: Please see attachment for suggestions, comments and questions. The data are of interest, but the wording of the manuscript needs improvement.

Reviewer #6: The authors have presented result of a study of Salmonella on chicken farms in Uganda. They report data on the distribution of serotypes, antimicrobial resistance profiles and plasmids. There are a number of interesting findings, but several issues that need to be addressed.

Lines 63-65 Sampling occurred over two seasons, but the time intervals for the dry and rainy seasons overlap. For samples collected from June- September, how was the season determined?

Line 66 It would be simpler just to say 2o chicken farms.

Lines 106-112 More description of the results would be useful. There were 20 chicken farms included. How many had positive samples? What was the distribution of serotypes by farm? Did some farms have a greater diversity than others? What was the distribution by season? Showing a significant p value, without presenting the underlying data is not useful. In line 107 it should be stated that the positive rate is for samples from chicken farms, not from chickens.

Line 113 There were multiple AMR phenotypes identified. Presenting the % of isolates resistant to each antibiotic is not that same as characterizing phenotypes.

Line 120 The sentence beginning “other than acquiring…” makes no sense as written.

Line 150 Although there is very likely a risk for Salmonella transmission from poultry in Uganda, the 13.5% positive rate of swab samples from chicken farms is not the same as a 13.5% prevalence of Salmonella on chicken carcasses or chicken meat samples at retail.

Line 158 As suggested above, there is need for more detail in the distribution of results by farm and season.

Table 1 An additional table that aggregates data by farm and season would be useful.

6. PLOS authors have the option to publish the peer review history of their article (what does this mean?). If published, this will include your full peer review and any attached files.

Reviewer #1: No

Reviewer #2: No

Reviewer #3: No

Reviewer #4: No

Reviewer #5: No

Reviewer #6: No

---

## [Author Response · Author response to Decision Letter 0]

9 Oct 2019

Reviewer #1: Review of International lineages of Salmonella enterica serovars isolated from chicken farms, Wakiso District, Uganda, by Ball et al.

In this manuscript, 51 isolates of Salmonella from chicken farms were characterized with respect to antimicrobial resistance, PCR testing and whole genome sequencing. The work seems to have been carried out carefully, but some details are missing in the materials and methods. Logistic regression was carried out with season as main factor, but it is not clear if only the total numbers of Salmonella strains were considered as response factor or also different genotypes and antimicrobial resistance factors. The main problem I have with the current manuscript is that differences were observed between seasons for total Salmonella counts, but it is not clear if the genotypic and phenotypic compositions of the Salmonella isolates were different between the rainy and dry seasons.

I made detailed comments in the manuscript itself; a scanned copy will be attached.

Title:

Replace Review of International Lineages by: Internationally recognized lineages.

This is not a review, and the lineages are all from Uganda. Completed

Abstract:

Generally ok,but see comments in the attachment. Addressed all comments in the attachment

Introduction:

OK, but add a paragraph about the methods of chicken raising in Uganda. Are the farms large-scale indoor chicken coops or are the chickens outdoors? This is important to be able to understand the differences between the rainy and dry seasons. Added information

Mention also about the movement of Salmonella between habitats (soil, feed, water, chickens, manure, floor, walls etc.) (Semenov et al., 2010). All housing was indoors.

Semenov, A.M., Kupriyanov, A.A. and van Bruggen, A.H.C. 2010. Transfer of enteric pathogens to successive habitats as part of microbial cycles. Microb. Ecol. 60: 239-249.

Methodology:

The rainy season and dry season overlap (line 64). Was the period of overlap between seasons avoided during sampling? Or were some samples considered in both categories? Corrected dates

The isolation method by Fedorka-Cray et al. was used; this is a classical method with the possibility that some colonies were considered Salmonella, but actually were not. How was the identity of all these isolates checked? By PCR? Which primers were used? That needs to be mentioned. See also Klerks et al., 2006.

Klerks, M.M., van Bruggen, A.H.C., Zijlstra, C., Donnikov M., and de Vos, R. 2006. Comparison of methods of extracting Salmonella enterica serovar Enteritidis DNA from environmental substrates and quantification of organisms by using a general internal procedural control. Appl. Environ. Microbiol. 72: 3879-3886. Added confirmation of methods

Results:

It is important to understand the differences between the rainy and dry seasons. I suggest indicating in Table 1 which samples were collected in each of the seasons. Then you can mention which serotypes occurred in the rainy season and which in the dry season. Now, the numbers may be too low in the dry season, but then it is important to mention that these are preliminary data, and that additional research would be needed to distinguish between seasons. Add rainy and dry season information in the table. Also added farm information. Added a second table to focus on season and serotypes

The outcome of the statistical analysis has not been presented in detail. I expect a multivariate graph, including all observations. Hopefully those will group out into two ‘clouds’, one for the rainy season and one for the dry season. The output provided for the stats was only the P-value.

Discussion:

To better understand the results, in particular the differences between dry and rainy seasons, you need to refer back to the way chickens are raised in Uganda (that’s why you need to add a paragraph about this in the introduction, so that you can refer to it in the discussion). Added information on the type of production Uganda uses

Which serotypes were endemic and which were introduced? Assumption made that all are endemic as recovery from wastewater and this study verify similarity. In the conclusion you mention endemic dissemination. In the rest of the manuscript, nothing was mentioned about dissemination. Are chickens imported or exported? Is the feed imported or exported? It is not really clear where all these serotypes and multi-drug-resistances came from. Or did the drug resistance develop in Uganda? It is important to give more general information about chicken production and the possible routes of dissemination of Salmonella. Origin of chickens is unknown. See additions in paper

Reviewer #2: The article by Ball et al. tackles the vital subject of antimicrobial resistance food-producing animals in a region from where very little is known regarding this subject. It is a welcome addition to the body of knowledge, although the article is missing information and data analysis.

1- The farms a poorly described both, in a geographical sense and relation to each other. Furthermore, the type of production, size, and information on different types of animals on the farm are not mentioned but are necessary. Added information on general area of farm location. We cannot supply geographical coordinates due to our IRB. Added information on other animals that were included on the farms and the type of production employed.

2- Spatial and temporal information on the tables is missing, and it is barely described in the results, and not discussed. Month of collection is described. Spatial location of farms is not possible

3- The presence of resistance genes without phenotypic testing is an indication of resistance, but the isolates cannot be described as resistant as it is the case in the discussion. 

This information is in the table under resistant profile (MIC). Isolates were tested phenotypically as described in the paper. We are unsure why the reviewer is querying.

5- Information on the quality of the sequencing obtained for each isolate should be documented. If they met Genometrackr standards, please estate this fact, and their current minimum. Samples are submitted to Genometrakr as mentioned in the methods section. The submission is indication that they are accepted by the program.

4-Exact information on the bioinformatics pipeline used to analyze the samples Is crucial for each isolate. Where different databases used every time? Where there any differences between the databases? Databases used added in the methods section.

Reviewer #3: This paper describes a cross sectional study of Salmonella in poultry farms in Uganda conducted over the dry and rainy seasons. The authors undertook antibiogram phenotyping of the isolate collection and used whole genome sequence analysis to determine the serotype, AMR genotype, plasmid replicon type and in some cases sequence type of the isolates. A high proportion of isolates were found to be MDR and to contain plasmids of significant impact to human health and alarmingly, fluoroquinolone resistance was very high (21.6%). Whilst the study design is excellent, the techniques used appropriate and the results are significant, the paper is let down by being poorly written in parts. Also, it is not stated whether the study population are broilers or layers (or mixed?). The chickens ranged from broilers, layers, Kuroilers and local birds were added I would also like to see more discussion about the very high rate of fluoroquinolone resistance and its relationship to antimicrobial use on farm. Added in the discussion. The only antibiotic used on most farms was oxytetracycline. The absence of resistance to third generation cephalosporins is also an important finding.

Specific comments below:

Line 45-47: This is a poor sentence that needs rewriting rewritten

Methods: Were these broiler farms or egg-producing farms or a combination of the two? This is a very important distinction due to the restrictions on which antimicrobials can be used in each system. The farm types were defined. Only Oxytet is used as described

Line 113: Tetracycline spelt incorrectly. It is not the AMR phenotype displaying resistance to eight antimicrobials but the isolate collection. Did not observe misspelling. Reworded the sentence

Line 120: Other than acquiring resistance genes does not make sense, This should be “Point mutations identified in the QRDR of fluoroquinolone target genes included……”

The section from Line 120 to Line 127 needs a careful rewrite as the sentence construction is poor. Rewritten

Line 128 to 130 is poorly written Rewritten

Reviewer #4: Multiple drug resistance (MDR) foodborne pathogens have been reported from multiple sources, especially from poultry worldwide. In this article, Ball et al. characterized 51 Salmonella enterica isolates, isolated from 400 environmental samples collected form poultry farms in Uganda by using whole genome sequencing (WGS). In addition, they also identified replicon plasmid through PCR. Among 51 isolates, 32 isolates were resistant to at least one antimicrobial and 10 isolates were assigned as MDR. Ball et al. also found a significant correlation between prevalence of Salmonella and seasonal difference. This article is underlining the importance of the surveillance systems, using One Health approach, to control MDR foodborne pathogens world wide. However, there are some minor issues that should be addressed and clarified before prior to publication.

Minor issues:

- Footnote, describing abbreviations of antimicrobials, should be added to Table 1. In addition, to describe temporal and spatial relatedness of isolates, new columns, representing where and when isolates were collected, might help. Added antimicrobials, added timeframe of collection. We cannot disclosed geographical coordinates but samples were collected in western rural and small town villages of the wakiso district, in Kampala Uganda. 

- For plasmid description, there are some inconsistent results, obtained from WGS and PCR. These results should be discussed in details. Both methods were used to identify plasmids and for the comparison. WGS identified plasmids that were identified in the PCR method, but also identified plasmids not in the PCR kit. 

- For antimicrobial resistance profiling, there are also some inconsistent results. For examples, some isolates (i.e., SALM-9, SALM-1) represented resistance by using MIC, but no gene, related to this phenotype, was found from WGS or vice versa. These results should be discussed in details. AMR phenotype only displays resistance when expressed whereas the gene could still be there. In the case of these isolates we observed cases when there was no resistance expressed but the gene was present. We also observed when the gene was not there but it was expressed phenotypically - this was interesting and needs more work as stated.

- In general, more discussion should be added.

Reviewer #5: Please see attachment for suggestions, comments and questions. The data are of interest, but the wording of the manuscript needs improvement. please see responses to attached suggestions

The manuscript would be more insightful if the authors would have been able to find out which antibiotics were used on the farms the samples came from. This was added in the manuscript. The only antibiotic used on the farms was oxytetracycline. 

Are there any data on which serovars are common in humans in Uganda? There is limited work looking at the common serotypes in humans. A part of this project was to do a parallel study in clinical samples with the MOH. But due to limited funding that part of the project was dropped until further funding. However, added in the paper was research on wastewater treatment plants and the findings of most common serotypes. 

Are the MDR levels higher or lower than those observed in other parts of the world? Because this study is a small-scale study, it is hard to compare to what is observed globally.

The manuscript contains numerous language issues and unclear sentences. Below are suggestions for improvements and questions that should be addressed. 

L.25: The growing occurrence of multidrug-resistant (MDR) Salmonella enterica in poultry is a public health concern worldwide. The present study, the prevalence, antimicrobial resistance, and sequence type [sequence of what?] of 51 Salmonella isolates recovered from 400 environmental samples from chicken farms in Uganda. reworded

L. 33: Five plasmid replicons were identified among all isolates, including IncFIIS 17/51 35 (33.3%), IncI1α 12/51 (23.5%), IncP 8/51 (15.7%), IncX1 8/51 (15.7%), and IncX2 1/51 (2.0%). In addition, ColpVC and IncFIB replicons were identified through WGS [spell out]. Spelled out WGS

L. 45: Additionally, international lineages [what are these?] have been readily spread …reworded

L. 48: In Uganda, antibiotics [what kind?] are increasingly being used [based on what information?] and not monitored or regulated in food-producing animals. Added information and citation

L. 50: To address this concern and considering the lack of information regarding antimicrobial resistance (AMR) in developing countries, Uganda ….reworded

L. 55: Therefore, we present a cross-sectional study involving chicken farms in Uganda to investigate the prevalence, AMR, and their?? [referring to what?] genomic aspects [what is meant by “aspects”?] of Salmonella enterica serovars. reworded

L. 60: Salmonella isolates were collected from chicken farms in parallel with the collection from cattle farms (5) as part of a cross-sectional study spanning one year. reworded

L. 76: Isolation of Salmonella was done as described by Fedorka-Cray et al. (8).reworded

L. 81: [Move this sentence to the previous section.] The isolates were frozen in LB broth with 30% glycerol (Thermo Fisher Scientific Inc, Waltham, MA) at -80o C. moved

L. 78: [Combine the sections.] Antimicrobial Resistance testing and molecular characterization of …[what?] Combined

A total of 51 Salmonella were obtained from chicken farms. For analyses, the isolates were retrieved from frozen stock, plated on Tryptic Soy Agar (TSA) with 5% sheep blood (BAP) (Thermo Fisher Scientific Inc, Waltham, MA) and incubated overnight at 37oC. AMR testing was done using the National Antimicrobial Resistance Monitoring System (NARMS) gram-negative panels (Thermo Fisher Scientific Inc, Waltham, MA) as described by Ball et al. (5). Lysates were prepared by suspending a loopful of well-isolated colonies in 200 µl of molecular grade water and vortexing at maximum speed for several seconds. The suspension was boiled at 100°C for 10 minutes, centrifuged at ..x g for 60 seconds, and the supernatant was collected for use as the DNA template. PCR screening [for what?] and whole genome sequencing were carried out as described in Ball et al. (manuscript submitted). [Details should be provided in case publication is delayed.]reworded

L. 96: Resistomes, plasmidomes and multilocus sequence types (MLST) were identified using ResFinder 3.1, PlasmidFinder 2.0, and MLST 2.0, respectively, available from the Center for Genomic Epidemiology (http://genomicepidemiology.org/). Sequence data were deposited in the GenomeTrakr Project. [I could not find any data on a project from Uganda. Please make sure it is available as you state.] Added the BIOsample number in table 1

L. 107: Fifty-one Salmonella were isolated (51/379; 13.5%) belonging eight different serovars: Enteritidis (31.3%); S. Kentucky (21.6%); S. Zanzibar and S. Virchow (15.7%); S. Newport and 42:r:- (5.88%), Typhimurium (4%) and S. Barranquilla (2.0%). The prevalence of Salmonella was higher in the rainy season (p=0.0017). The isolates displayed resistance to eight antimicrobials including tetracylcine (51%), …. reworded

L. 120: Other than acquiring resistance genes???[I do not understand this part of the sentence.] were assigned as quinolone resistance determining regions (QRDR) with point mutation in gyrA and parC (Table 1). Ten isolates (19.6%) showed a double amino acid mutation in GyrA (GyrA-S83F-D87N), whereas eight isolates (15.6%) showed a single amino acid substitution of serine to tyrosine at codon 126. For QRDR in parC (n=10; 19.6%) only one substitution of serine by isoleucine at codon 80 was observed. No mutations were found in gyrB and parE. Sequencing identified six plasmids. IncFII(S)-IncFIB (S)- ColpVC were the most common in S. Enteritidis; Incl1-ColpVC in S. Kentucky and S. Zanzibar; IncX2 in S. Newport; Incl1-IncFII(S)-IncFIB (S)-ColpVC in S. Typhimurium and Col440I in serovar 42:r:-. Nine sequence types, namely ST11, ST198, ST466, ST16, ST166, ST46, ST19, ST1208 and ST3807 were associated with S. Enteritidis, S. Kentucky, S. Zanzibar, 135 S. Virchow, S. Newport, S. Newport, S. Typhimurium, S. serovar 42:r:- and S. 136 Barranquilla, respectively. reworded

L. 150: The percent prevalence of Salmonella (13.5%) in this study highlights the potential risk to humans to Ugandan households. This percentage is higher than that reported by Afema et al. (9) of 6.6% Salmonella in samples from live birds markets within Kampala, Uganda. [How meaningful is this comparison? Farms vs. live birds?] [Explain why the percentage might be higher.] reworded

There was a seasonal effect in the recovery of Salmonella. Uganda typically has a rainy season that occurs between March to May and October to December (10). Recovery of Salmonella was higher during the rainy season, possibly because the higher humidity and moisture allowed better dispersal or survival of Salmonella as observed for other bacterial species in poultry (11). Comparable to the situation in the US (12), Salmonella serovars Enteritidis and Kentucky were most often recovered. Serovar Kentucky has previously been reported in Uganda in humans, poultry, and the environment (9). reworded

Among chicken isolates, Salmonella presented with MDR phenotypes to the 168 antimicrobials tested. [New paragraph] Approximately 38% of the isolates were resistant to two or more classes of antimicrobials? [antibiotics?], including two isolates resistant to seven antimicrobials. The Salmonella serovar Kentucky isolates were resistant to over five (ciprofloxacin, nalidixic acid, streptomycin, sulfisoxazole, and tetracycline) or seven (chloramphenicol, ampicillin, ciprofloxacin, nalidixic acid, streptomycin, sulfisoxazole, tetracycline, and trimethoprim-sulfamethoxazole) antibiotics. All Salmonella serovar Kentucky isolates were resistant to ciprofloxacin. Since the early 2000s, ciprofloxacin resistant Salmonella serovar Kentucky have been on the rise, especially in travelers to northern and eastern Africa (13). Rickert-Hartman et al. (..??) found that 9% of the Salmonella serovar Kentucky isolated from travelers were ciprofloxacin resistant. Poultry was thought to be a reservoir for these resistant strains (13, 14). Ciprofloxacin-resistant Kentucky have caused seven persons to become ill and one death in the US after the carriers had travelled to India (13). In this regard, the emergence of S. Kentucky ST198 poses a major threat to public health worldwide since it is highly drug-resistant (15) and has been reported in different sources including retail chicken carcasses (16). reworded

The presence of mutation can be useful for tracking the pandemic ciprofloxacin-resistant S. Kentucky strain ST198 from geographically distinct regions (15). Using WGS TEM-1B (beta lactamase?) was identified in five isolates that PCR methods did not identify. In previous studies (17), discrepancies were also seen between phenotypic resistance and genotypic analysis using WGS. It was reported that an MIC might not reach the breakpoint even though resistance genes were present (17). Five of the 28 plasmids that were screened through PCR were observed in multiple isolates: IncFIIS (17/51; 33.3%), IncI1α (12/51; 23.5%), IncP (8/51; 15.7%), 193 IncX1 (8/51; 15.7%), and IncX2 (1/51; 2.0%). After analyzing the WGS sequences for plasmids, 12 isolates were found to harbor IncI1α, with seven of the 12 having an additional plasmid (ColpVC) that was not detected by PCR and two with IncFIIS plasmid. Seventeen isolates carried the IncFIIS plasmid. These same 17 isolates also presented IncFIB (S) plasmids, and ColpVC and Col4401 were identified in seven and two isolates, respectively. IncX2 and IncP were not identified in the WGS analysis and by PCR. PCR did not detect plasmids in ten isolates, but WGS detected ColpVC 201 in nine isolates and Col4401 in one isolate. IncFIIS was the most common plasmid identified at 33.3% (17/51). Studies have shown that isolates containing blaCTX-M-1 harbor IncFIIS along with other incompatibility plasmids (18). Inc1 plasmids are known to be distributed throughout many serotypes of Salmonella and predominate in both E. coli and Salmonella 206 (19-21). In this study, Inc1α was observed among Salmonella serovars Zanzibar, Kentucky, and Typhimurium. All isolates from Salmonella serovar Kentucky and Typhimurium came from the same farm [what does this observation suggest?]. IncP and IncX1 were the next most common plasmids detected by PCR. Both were present in the Salmonella serovar Virchow isolates. It has been reported that IncP plasmids can spread via conjugative transfer and that they code for a broad range antimicrobial resistances. IncP is highly likely to be found in manure, wastewater, and soil (22). IncX1 is commonly found as a narrow host-range plasmid in Enterobacteriaceae, also spreading to other bacteria via conjugative transfer (23). reworded

L. 216: In summary, we present in this study the clonal??[how do you know this?] distribution of eight Salmonella enterica serovars displaying resistance to clinically important antibiotics. Of these, the presence of international lineages such as ciprofloxacin-resistant S. Kentucky sequence type 198 in chicken farms raises a public concern given that fluoroquinolones are the first treatment choice. Our findings suggest that endemic dissemination of resistant serovars …[incomplete sentence], adding valuable information to the epidemiological surveillance in Uganda. Therefore, these results may encourage addition genomic surveillance studies in this region to aid in the development of mitigation strategies to limit the global distribution of these multidrug resistant Salmonella enterica. reworded

Reviewer #6: The authors have presented result of a study of Salmonella on chicken farms in Uganda. They report data on the distribution of serotypes, antimicrobial resistance profiles and plasmids. There are a number of interesting findings, but several issues that need to be addressed.

Lines 63-65 Sampling occurred over two seasons, but the time intervals for the dry and rainy seasons overlap. For samples collected from June- September, how was the season determined? Corrected the dates

Line 66 It would be simpler just to say 2o chicken farms. reworded

Lines 106-112 More description of the results would be useful. There were 20 chicken farms included. How many had positive samples? What was the distribution of serotypes by farm? Did some farms have a greater diversity than others? What was the distribution by season? Showing a significant p value, without presenting the underlying data is not useful. In line 107 it should be stated that the positive rate is for samples from chicken farms, not from chickens. Please see an updated table one and table 2 for this information

Line 113 There were multiple AMR phenotypes identified. Presenting the % of isolates resistant to each antibiotic is not that same as characterizing phenotypes. Added table 3 and 4 to show more on AMR phenotype 

Line 120 The sentence beginning “other than acquiring…” makes no sense as written. reworded

Line 150 Although there is very likely a risk for Salmonella transmission from poultry in Uganda, the 13.5% positive rate of swab samples from chicken farms is not the same as a 13.5% prevalence of Salmonella on chicken carcasses or chicken meat samples at retail. reworked. 

Line 158 As suggested above, there is need for more detail in the distribution of results by farm and season. Information was added in table 1

Table 1 An additional table that aggregates data by farm and season would be useful. Tabel 2 was added to show this information.

---

## [Decision Letter · Decision Letter 1]

31 Oct 2019

PONE-D-19-19750R1

International lineages of Salmonella enterica serovars isolated from chicken farms, Wakiso District, Uganda

PLOS ONE

Dear Dr. Ball,

Thank you for submitting your manuscript to PLOS ONE. After careful consideration, we feel that it has merit but does not fully meet PLOS ONE’s publication criteria as it currently stands. Therefore, we invite you to submit a revised version of the manuscript that addresses the points raised during the review process.

We would appreciate receiving your revised manuscript by Dec 15 2019 11:59PM. To enhance the reproducibility of your results, we recommend that if applicable you deposit your laboratory protocols in protocols.io, where a protocol can be assigned its own identifier (DOI) such that it can be cited independently in the future. For instructions see: http://journals.plos.org/plosone/s/submission-guidelines#loc-laboratory-protocols

We look forward to receiving your revised manuscript.

Kind regards,

Feng Gao

Academic Editor

PLOS ONE

Reviewers' comments:

Reviewer's Responses to Questions

**Comments to the Author**

1. If the authors have adequately addressed your comments raised in a previous round of review and you feel that this manuscript is now acceptable for publication, you may indicate that here to bypass the “Comments to the Author” section, enter your conflict of interest statement in the “Confidential to Editor” section, and submit your "Accept" recommendation.

Reviewer #3: All comments have been addressed

Reviewer #5: (No Response)

Reviewer #6: (No Response)

2. Is the manuscript technically sound, and do the data support the conclusions?

Reviewer #3: Yes

Reviewer #5: Yes

Reviewer #6: Partly

3. Has the statistical analysis been performed appropriately and rigorously? 

Reviewer #3: Yes

Reviewer #5: I Don't Know

Reviewer #6: No

4. Have the authors made all data underlying the findings in their manuscript fully available?

Reviewer #3: Yes

Reviewer #5: (No Response)

Reviewer #6: No

5. Is the manuscript presented in an intelligible fashion and written in standard English?

Reviewer #3: Yes

Reviewer #5: Yes

Reviewer #6: Yes

6. Review Comments to the Author

Reviewer #3: (No Response)

Reviewer #5: The manuscript has been improved considerably. The one comment I have now that Table 1 is available concerns n = 51. Is it possible that you included the same strains multiple times? The strains could have been isolated from different sites at a farm or even at different times. Would inclusion of the same strains not skew the percentages of resistance to particular antibiotics?

Did the whole genome sequencing effort clearly demonstrate that all 51 isolates were different?

Reviewer #6: Authors have revised paper in response to previous comments, but still need to address concerns. The over point of the study is to highlight the need for integrated surveillance for Salmonella using a One Health approach. I certainly concur with that, but think there is considerable restructuring of the paper needed to draw out that point.

In this relatively small data set, all of the MDR strains were S. Kentucky ST 198, which likely represent the expansion of a globally emerging strain. The authors note that these were from one farm that did not use fluoroquinolones. The spread of emerging strains across borders is a critical reason for integrated surveillance.

Specific comments:

Line 34 delete “all”

Line 61 does “chicken” imply live animals or poultry meat? Please clarify.

Line 106 delete “all”

Line 139-140 It is important to expand on this analysis. 6/20 farms were positive in the dry season and 9/18 were positive during the wet season. The rate of Salmonella positivity appears to be about 9% for dry season, and 19% for rainy season. These are the data that are more interesting than the p-value that should be presented.

Line 142-145 It should be noted that all SE were pansusceptible, and all MDR isolates were S. Kentucky. The resistance data follow clonal strains of Salmonella.

Line 145 Table 2 does not include resistance data.

Line 170 Note ass are S. Kentucky

Line 209-213. See comments for lines 139-140. There is no point to mentioning the logistic regression if none of the output of the regression is presented.

Lines 182-208 There is a lot of detail presented here, but the context is not clear, and the reader easily gets lost. What is the bottom line? This is an opportunity to show how WGS can aid surveillance, but that does not clearly emerge from the presentation.

Line 237-239 Does this mean that surveillance data for human illnesses were compared to results of this study? If so, that should be more thoroughly described and discussed. It would be useful to do so to highlight the One Health approach.

Line 258 and beyond… The presence of MDR Kentucky likely represents movement of clonal strains. This is the good justification of One Health integrated surveillance.

7. PLOS authors have the option to publish the peer review history of their article (what does this mean?). If published, this will include your full peer review and any attached files.

Reviewer #3: No

Reviewer #5: Yes: Rolf Joerger

Reviewer #6: No

---

## [Author Response · Author response to Decision Letter 1]

17 Dec 2019

Response to editors, Manuscript PONE-D-19-19750.R2

6. Review Comments to the Author

Response: We sincerely appreciate the editorial board and reviewers for their careful review and constructive suggestions. We believe that the manuscript was substantially improved after making the suggested edits. We have responded to specific queries below.

Reviewer #3: (No Response)

Reviewer #5: The manuscript has been improved considerably. 

Response: We sincerely thank the reviewer for their favorable comments and interest in our work.

The one comment I have now that Table 1 is available concerns n = 51. Is it possible that you included the same strains multiple times? The strains could have been isolated from different sites at a farm or even at different times. Would inclusion of the same strains not skew the percentages of resistance to particular antibiotics?

Response: There were isolates collected on the same farms but at different sample sites (feed, water, litter, entryway, etc.) and different months. Most farms had new flocks in the area or in the process of getting new flocks between visits. The object of the entire study (E. coli and Salmonella for Cattle and Chicken) was to see if the prevalence and AMR results differ between the seasons and to identify what would be found. Every strain that was collected was presented in Table one to see all phenotypic and genotypic differences in both seasons. To avoid the percentages skew, we did not include the same strains multiple times, as mentioned above.

Did the whole genome sequencing effort clearly demonstrate that all 51 isolates were different?

Response: Although traditional tools have been considered the gold standards to study Salmonella, the whole-genome sequencing has been applied as an alternative to that after its establishment. In this regard, WGS recognizes antimicrobial resistance profile, MLST and evolutionary groupings that could precisely identify the differences between Salmonella strains. In this backdrop, we observed that the main drivers for cluster analysis were serotype, sequence types, and resistance profile since all isolates were clustered together by these characteristics and not by the period of isolation, source, or geographic location. We do have PFGE results not included in this manuscript for these isolates. They were not included due to the inclusion of other strains, not in this study. However, these PFGE results are in line with the WGS sequence data. 

Reviewer #6: Authors have revised paper in response to previous comments, but still need to address concerns. The over point of the study is to highlight the need for integrated surveillance for Salmonella using a One Health approach. I certainly concur with that, but think there is considerable restructuring of the paper needed to draw out that point.

In this relatively small data set, all of the MDR strains were S. Kentucky ST 198, which likely represent the expansion of a globally emerging strain. The authors note that these were from one farm that did not use fluoroquinolones. The spread of emerging strains across borders is a critical reason for integrated surveillance.

Specific comments:

Line 34 delete “all” 

Response: Addressed

Line 61 does “chicken” imply live animals or poultry meat? Please clarify. 

Response: Live birds. Addressed

Line 106 delete “all” 

Response: Addressed

Line 139-140 It is important to expand on this analysis. 6/20 farms were positive in the dry season and 9/18 were positive during the wet season. The rate of Salmonella positivity appears to be about 9% for dry season, and 19% for rainy season. These are the data that are more interesting than the p-value that should be presented.

Response: Remove the logistic regression statement. We agree that it is essential to expand this study. Indeed, these results may serve as a baseline study for future surveys. 

Line 142-145 It should be noted that all SE were pansusceptible, and all MDR isolates were S. Kentucky. The resistance data follow clonal strains of Salmonella. 

Response: Addressed

Line 145 Table 2 does not include resistance data.

Response: Addressed

Line 170 Note ass are S. Kentucky

Response: Addressed

Line 209-213. See comments for lines 139-140. There is no point to mentioning the logistic regression if none of the output of the regression is presented. 

Response: Addressed

Lines 182-208 There is a lot of detail presented here, but the context is not clear, and the reader easily gets lost. What is the bottom line? This is an opportunity to show how WGS can aid surveillance, but that does not clearly emerge from the presentation. 

Response: Statement was included “Although traditional tools have been considered the gold standard to study Salmonella, WGS has been applied as an alternative in providing more detailed and accurate data. In this regard, WGS recognizes antimicrobial resistance profile, MLST, and evolutionary groupings that could precisely identify the differences between Salmonella strains. We observed that the main drivers for characterization analysis were serotype, sequence types, and resistance profile. These isolates were clustered together by these characteristics and not by a period of isolation, source, or geographic location. To endorse these results, we have done pulsed-field gel electrophoreses (results not included), which are in agreement with the WGS results. Our study shows how WGS inspection constitutes a useful means to characterize Salmonella isolates.”

Line 237-239 Does this mean that surveillance data for human illnesses were compared to results of this study? If so, that should be more thoroughly described and discussed. It would be useful to do so to highlight the One Health approach. 

Response: There were no human results compared in parallel to this study. The study mentioned in the manuscript was a project in Uganda looking at a few human cases, not for surveillance. The idea for our study was to get human data in parallel to compare, but resources were limited to do so. The statement in this line was implying that we did not see the same type of serovars as they saw in the Afema article. Clarification was added 

Line 258 and beyond… The presence of MDR Kentucky likely represents movement of clonal strains. This is the good justification of One Health integrated surveillance. 

Response: Addressed

---

## [Decision Letter · Decision Letter 2]

7 Jan 2020

International lineages of Salmonella enterica serovars isolated from chicken farms, Wakiso District, Uganda

PONE-D-19-19750R2

Dear Dr. Ball,

We are pleased to inform you that your manuscript has been judged scientifically suitable for publication and will be formally accepted for publication once it complies with all outstanding technical requirements.

With kind regards,

Feng Gao

Academic Editor

PLOS ONE

Additional Editor Comments (optional):

Reviewers' comments:

Reviewer's Responses to Questions

**Comments to the Author**

1. If the authors have adequately addressed your comments raised in a previous round of review and you feel that this manuscript is now acceptable for publication, you may indicate that here to bypass the “Comments to the Author” section, enter your conflict of interest statement in the “Confidential to Editor” section, and submit your "Accept" recommendation.

Reviewer #5: All comments have been addressed

Reviewer #6: All comments have been addressed

2. Is the manuscript technically sound, and do the data support the conclusions?

Reviewer #5: Yes

Reviewer #6: (No Response)

3. Has the statistical analysis been performed appropriately and rigorously? 

Reviewer #5: I Don't Know

Reviewer #6: (No Response)

4. Have the authors made all data underlying the findings in their manuscript fully available?

Reviewer #5: Yes

Reviewer #6: (No Response)

5. Is the manuscript presented in an intelligible fashion and written in standard English?

Reviewer #5: Yes

Reviewer #6: (No Response)

6. Review Comments to the Author

Reviewer #5: My concerns have been addressed to the extent possible. The limitations of the study are now clearer.

Reviewer #6: (No Response)

7. PLOS authors have the option to publish the peer review history of their article (what does this mean?). If published, this will include your full peer review and any attached files.

Reviewer #5: Yes: Rolf D. Joerger

Reviewer #6: No

---

## [Editor Report · Acceptance letter]

15 Jan 2020

PONE-D-19-19750R2 

International lineages of *Salmonella enterica* serovars isolated from chicken farms, Wakiso District, Uganda 

Dear Dr. Ball:

I am pleased to inform you that your manuscript has been deemed suitable for publication in PLOS ONE. Congratulations! Your manuscript is now with our production department. 

With kind regards,

on behalf of

Dr. Feng Gao 

Academic Editor

PLOS ONE